

# Evaluation of global fire simulations in CMIP6 Earth system models

Fang Li[1], Xiang Song[1], Sandy P. Harrison[2], Jennifer R. Marlon[3], Zhongda Lin[4], L. Ruby Leung[5], Jörg Schwinger[6], Virginie Marécal[7], Shiyu Wang[8], Daniel S. Ward[9], Xiao Dong[1], Hanna Lee[10], Lars Nieradzik[11], Sam S. Rabin[12], Roland Séférian[7]

[1] International Center for Climate and Environment Sciences, Institute of Atmospheric Physics, Chinese Academy of Sciences, Beijing, 100029, China

[2] Department of Geography and Environmental Science, University of Reading, Reading, RG6 6AB, UK

[3] School of the Environment, Yale University, New Haven, CT 06511, USA

[4] State Key Laboratory of Numerical Modeling for Atmospheric Sciences and Geophysical Fluid Dynamics, Institute of Atmospheric Physics, Chinese Academy of Sciences, Beijing, 100029, China

[5] Atmospheric, Climate and Earth Sciences Division, Pacific Northwest National Laboratory, Richland, WA, USA

[6] NORCE Norwegian Research Centre & Bjerknes Centre for Climate Research, Bergen, Norway

[7] Centre National de Recherches Météorologiques, Université de Toulouse, Météo-France, CNRS, Toulouse, 31000, France

[8] Swedish Meteorological and Hydrological Institute (SMHI), Norrköping, 60176, Sweden

[9] Karen Clark and Company, Boston, MA, USA

[10] Department of Biology, Norwegian University of Science and Technology, Trondheim, Norway

[11] Department of Physical Geography and Ecosystem Science, Lund University, Lund, Sweden

[12] Climate and Global Dynamics Laboratory, National Center for Atmospheric Research, Boulder, Colorado 80305, USA

*Correspondence to*: Fang Li (lifang@mail.iap.ac.cn)

**Abstract** Fire is the primary form of terrestrial ecosystem disturbance on a global scale and an important Earth system process. Most Earth system models (ESMs) have incorporated fire modeling, with 19 out of them submitting model outputs of fire-related variables to the Coupled Model Intercomparison Project Phase 6 (CMIP6). This study provides the first comprehensive evaluation of

5    CMIP6 historical fire simulations by comparing them with multiple satellite-based products and charcoal-based historical reconstructions. Our results show that most CMIP6 models simulate the present-day global burned area and fire carbon emissions within the range of satellite-based products. They also capture the major features of observed spatial patterns and seasonal cycles, the relationship of fires with precipitation and population density, and the influence of El Niño-Southern Oscillation

10   (ENSO) on the interannual variability of tropical fires. Regional fire carbon emissions simulated by the CMIP6 models from 1850 to 2010 generally align with the charcoal-based reconstructions, although there are regional mismatches, such as in southern South America and eastern temperate North



America prior to the 1910s and in temperate North America, eastern boreal North America, Europe, and boreal Asia since the 1980s. The CMIP6 simulations have addressed three critical issues identified in the CMIP5: (1) the simulated global burned area less than half of the observations, (2) the failure to reproduce the high burned area fraction observed in Africa, and (3) the weak fire seasonal variability.

Furthermore, the CMIP6 models exhibit improved accuracy in capturing the observed relationship between fires and both climatic and socioeconomic drivers, and better align with the historical long-term trends indicated by charcoal-based reconstructions in most regions worldwide. However, the CMIP6 models still fail to reproduce the decline in global burned area and fire carbon emissions observed over the past two decades, mainly attributed to an underestimation of anthropogenic fire

suppression, and the spring peak in fires in the Northern Hemisphere mid-latitudes, mainly due to an underestimation of crop fires. In addition, the model underestimates the fire sensitivity to wet-dry conditions, indicating the need to improve fuel wetness estimation. Based on these findings, we present specific guidance for fire scheme development and suggest the post-processing methodology for using CMIP6 multi-model outputs to generate reliable fire projection products.

## 1 Introduction

Fire is the primary form of terrestrial ecosystem disturbance on a global scale and a critical Earth system process (Randerson et al., 2006; Bowman et al., 2009). Fire has occurred since the emergence of terrestrial plants over 400 million years ago (Scott and Glasspool, 2006; Bowman et al., 2009), and

presently burns more than 400 Mha of vegetated land and emits 2−3 Pg carbon globally each year (van der Werf et al., 2017; Giglio et al., 2018; Chuvieco et al., 2018; Chen et al., 2023). Fire is regulated by climate and weather, vegetation characteristics, and human activities, and at the same time, influences them in multiple ways, resulting in intricate feedback loops (Bond-Lamberty et al., 2009; Jiang et al., 2016; Li and Lawrence, 2017; Li et al., 2017, 2019; Lasslop et al., 2020; Kim et al., 2020; Wu et al.,

2022; Lou et al., 2023). Despite a reduction in the global burned area over the past two decades, emissions from forest fires and the occurrence of extreme fires have increased (Andela et al., 2017; Zheng et al., 2021). Moreover, global fires are projected to rise in most regions of the world, particularly if climate mitigation efforts are weak (Li et al., 2021; Yu et al., 2022; UNEP, 2022).



Earth system models (ESMs) simulate the processes and interactions within and across the atmosphere, land, ocean, sea ice, and biosphere, which are crucial for analyzing historical climate and environmental changes and for projecting the Earth's future (Scholze et al., 2012; Danabasoglu et al., 2020; Song et al., 2021). ESMs became the predominant coupled model type in the Coupled Model

Intercomparison Project Phase 6 (CMIP6; Eyring et al., 2016), which is the latest iteration of CMIP to release model outputs for general use, and supports the IPCC AR6 (IPCC, 2021). Given the critical role of fire in the Earth system, most ESMs already include fire modeling.

Kloster and Lasslop (2017) assessed fire simulations in CMIP5, based on nine models that had submitted historical fire simulations. They found these models severely underestimated the global

burned area by more than 50% compared to observations, although the simulated global fire carbon emissions were within the range of observations. They also showed that all CMIP5 models failed to reproduce the spatial patterns of the burned area mainly because they underestimated the high values in Africa, and only MPI-ESM performed better than a random model in simulating the observed seasonal phase of burned area.

Many more models have conducted fire-enabled historical simulations for CMIP6, in which the most used fire scheme has evolved from GlobFIRM (Thonicke et al., 2001) in CMIP5 to the Li scheme (Li et al., 2012, 2013; Li and Lawrence, 2017). However, it remains unknown how well CMIP6 ESMs perform in fire simulations. This study provides the first comprehensive evaluation of CMIP6 fire simulations, including the global total, spatial pattern, seasonality, recent and historical trends, and

interannual variability of burned area and fire carbon emissions. To disentangle biases and inter-model differences arising from fire parameterization schemes and climate simulations, we also evaluate the modeled relationship between fires and two key driving variables: precipitation in the tropics and subtropics (35ºS to 35ºN) and population density globally. This evaluation can deepen our understanding of past, present, and future changes in fires, as well as the closely related carbon cycle, within CMIP6

simulations. Based on the results, we also suggest strategies for fire scheme development and for the post-processing methodology of CMIP6 multi-model ensemble simulations to generate more reliable projections of future fire changes.

**2 Data and methods**



**2.1 Fire simulations**

We downloaded CMIP6 historical fire simulations that cover the period of 1850−2014 from http://esgf-node.llnl.gov/search/cmip6/ (last accessed: March 2023) (Eyring et al., 2016). 19 ESMs submitted fire simulations, of which 9 models submitted burned area and 18 models submitted fire carbon emissions

(Table 1). All the simulations were driven by the same forcing data, e.g., prescribed greenhouse gas concentration (Meinshausen et al., 2017), anthropogenic and biomass burning emissions (Feng et al., 2020), and land use and land cover change (Hurtt et al., 2020).

**Table 1.** Summary description of CMIP6 ESMs used in the study.

| ESMs | Institute | BA | Fire C | Land model | Fire scheme | Human ign / sup | Crop fires |
|---|---|---|---|---|---|---|---|
| AWI-ESM-1-1-LR[1] | AWI (Germany) | | √ | JSBACH3.2 | SPITFIRE[d] (modified) | √ / √ | 0 |
| CESM2[2] | NCAR (USA) | √ | √ | CLM5 | Li[a] | √ / √ | √ |
| CESM2-WACCM[2] | NCAR (USA) | √ | √ | CLM5 | Li[a] | √ / √ | √ |
| CMCC-CM2-SR5[3] | CMCC (Italy) | √ | √ | CLM4.5 | Li[b] | √ / √ | √ |
| CMCC-ESM2[4] | CMCC (Italy) | √ | √ | CLM4.5 | Li[b] | √ / √ | √ |
| CNRM-ESM2-1[5] | CNRM-CERFACSE (France) | √ | √ | ISBA-CTRIP | GlobFIRM[e] (modified) | / √ | 0 |
| E3SM-1-1[6] | DOE (USA) | | √ | ELM | Li[b] | √ / √ | √ |
| E3SM-1-1-ECA[6] | DOE (USA) | | √ | ELM | Li[b] | √ / √ | √ |
| EC-Earth3-CC[7] | EC-Earth-Cons. (Europe) | √ | √ | LPJ-GUESS | GlobFIRM | | |
| EC-Earth3-Veg[7] | EC-Earth-Cons. (Europe) | √ | √ | LPJ-GUESS | GlobFIRM | | |
| EC-Earth3-Veg-LR[7] | EC-Earth-Cons. (Europe) | | √ | LPJ-GUESS | GlobFIRM | | |
| GFDL-ESM4[8] | NOAA-GFDL (USA) | | √ | LM4.1 | FINAL[f] Li[c] | √ / √ | √ |
| MPI-ESM1-2-HAM[9] | HAMMOZ-Cons. (Europe) | | √ | JSBACH3.2 | SPITFIRE[d] (modified) | √ / √ | 0 |
| MPI-ESM1-2-LR[10] | MPI (Germany) | | √ | JSBACH3.2 | SPITFIRE[d] (modified) | √ / √ | 0 |
| MRI-ESM2-0[11] | MRI (Japan) | | √ | HAL1 | GlobFIRM | | |
| NorCPM1[12] | NCC (Norway) | | √ | CLM4 | GlobFIRM | | |
| NorESM2-LM[13] | NCC (Norway) | √ | √ | CLM5 | Li[b] | √ / √ | √ |
| NorESM2-MM[13] | NCC (Norway) | √ | √ | CLM5 | Li[b] | √ / √ | √ |
| TaiESM1-0[14] | AS-RCEC (Taiwan, China) | | √ | CLM4 | GlobFIRM[g] | √ / √ | |

[a] Li et al. (2012, 2013) and Li and Lawrence (2017); [b] Li et al. (2012, 2013); [c] Li et al. (2012); [d] SPITFIRE (Thonicke et al., 2010) with modifications from Lasslop et al. (2014); [e] GlobFIRM (Thonicke et al., 2001), but adapting to a daily timestep, tuning parameters, and assuming no fire in grid cell where cropland fraction is over 20%; [f] FINAL: Li et al. (2012) but tuning parameters and using prescribed cropland and pasture fires based on GFED3 (Rabin et al., 2018) as well as introducing

the landscape fragmentation effect on fire spread, multiday burning, and SPITFIRE canopy fire scheme (Ward et al., 2018); [g] GlobFIRM (Thonicke et al., 2001) but adapted to a sub-hour timestep (Kloster et al., 2010) ; references for ESMs: [1] Contzen et al. (2022); [2] Danabasoglu et al. (2020); [3] Cherchi et al. (2019); [4] Lovato et al. (2022); [5] Séférian et al. (2019); [6] Burrows et al. (2020); [7] Döscher et al. (2022); [8]





Dunne et al. (2020); [9] Neubauer et al. (2019); [10] Mauritsen et al. (2019); [11] Yukimoto et al. (2019); [12] Bethke et al. (2021); [13] Seland et al. (2020); [14] Lee et al. (2020).

The fire schemes employed in all 19 ESMs are process-based, simulating both the processes of fire occurrence and fire spread. Of the 9 models providing burned area data, 6 used the Li scheme (Li et al., 2012, 2013; Li and Lawrence, 2017), while the remaining 3 utilized the GlobFIRM scheme (Thonicke et al., 2001). Among the 18 models that provided fire carbon emissions data, 8 adopted the Li scheme, 7 employed the GlobFIRM, and 3 used the modified SPITFIRE scheme (Thonicke et al., 2010) by Lasslop et al. (2014). The SPITFIRE scheme is the most complex since it uses the Rothermel model to calculate the fire spread rate in the downwind direction, considers the impact of the fuel structure, and distinguishes surface and canopy fires.

The fire schemes differ in their fundamental equations for calculating the burned area. The Li fire scheme and SPITFIRE calculate the time-step area burned in a grid cell as a product of the number of fires and the average spread area per fire. For GlobFIRM, the annual burned area fraction is a nonlinear function of fire season length, in which the fire season length is calculated by summing fire occurrence probability throughout the year. CNRM-ESM2-1 modifies the annual calculation of GlobFIRM to a daily time step using the methodology of Krinner et al. (2005) for simulations of fire seasonality (Delire et al., 2020).

The fire schemes also vary in how they model the anthropogenic influence on fires (Table 1). GlobFIRM does not account for direct human effects on fires, but its variant (used in CNRM-ESM2-1) considers human suppression by assuming no fire occurrence when croplands cover more than 20% of the grid cell. The Li scheme models crop fires, fires caused by anthropogenic deforestation in tropical closed forests, and human ignition and suppression of both fire occurrence and spread in regions outside of tropical closed forests and croplands. However, in the CESM2, CESM2-WACCM, NorESM2-LM, and NorESM2-MM simulations for CMIP6, crop module was active and assumed no fires occurred in managed croplands. The variant of SPITFIRE used in MPI-ESM1-2-HAM, MPI-ESM1-2-LR, and MPI-ESM2-0 also considers human ignition and suppression on fire occurrence, and sets burned area to zero in croplands. In addition, all the ESMs treat fires in pasturelands as natural grassland fires, except for GFDL-ESM4.1, which uses prescribed pasture fires derived from the multi-year average burned area of Global Fire Emissions Database version 3 with small fires (GFED3s)



(Rabin et al., 2018). The MPI-ESM family (using a variant of SPITFIRE) set high fuel bulk density for pasture plant functional type (PFT), which indirectly distinguishes these from the natural grassland fires due to differences in fuel availability.

The fire schemes calculate fire carbon emissions by multiplying the burned area, fuel load, and combustion completeness. The combustion completeness is a proportion (0–100 %) of live plant tissues and ground litter consumed by fires. It depends on PFT and plant tissue type in both the GlobFIRM and Li schemes, and on fuel type and wetness in SPITFIRE.

For comparison, we downloaded CMIP5 historical fire simulations from http://esgf-node.llnl.gov/search/cmip5/ (last accessed: March 2023). The CMIP5 historical simulations cover the

period from 1850 to 2005 (Taylor et al., 2012), and thus end 9 years earlier than CMIP6 historical simulations. Seven models submitted burned area simulations in CMIP5 (CCSM4, CESM1-BGC, CESM1-CAM5, CESM1-FASTCHEM, CESM1-WACCM, MPI-ESM-LR, and MPI-ESM-MR) and 12 models submitted fire carbon emissions simulations (BNU-ESM, CCSM4, CESM1-BGC, CESM1-FASTCHEM, CESM1-WACCM, CMCC-CESM, GFDL-ESM2G, GFDL-ESM2M, IPSL-CM5-LR,

IPSL-CM5-MR, MPI-ESM-LR, MPI-ESM-MR). The majority of the CMIP5 models (5 out of 7 and 8 out of 12) used the GlobFIRM fire scheme (Thonicke et al., 2001), but adapted the scheme's annual timestep to a sub-daily to monthly timestep using the similar method of CNRM-ESM2-1 in CMIP6 (Krinner et al., 2005; Kloster et al., 2010, 2017).

### 2.2 Fire benchmarks

There are differences between satellite-based fire products (Li et al., 2019; Hantson et al., 2020). To account for the uncertainty in observations, we employed multiple products as benchmarks.

For burned area, we used the Global Fire Emissions Database version 5 (GFED5; Chen et al., 2023), the European Space Agency Fire Climate Change Initiative version 5.1 (FireCCI51; Chuvieco et al., 2018), and the Collection 6 Moderate Resolution Imaging Spectroradiometer (MODIS C6; Giglio et

al., 2018), all of which provide monthly data at 0.25° spatial resolution. We used the period of 2001−2014 to compare with the CMIP6 historical simulations. Burned area since 2001 in GFED5 is based on the MODIS global burned area product MCD64A1, with omission and commission errors corrected by dynamic adjustment factors estimated using the Landsat or Sentinel-2 burned area (Chen et al., 2023). The FireCCI51 burned area is derived using the MODIS C6 250-m daily surface



reflectance, MCD14ML 1-km daily active fire products, and a two-phase approach for seed detection and regional growth (Chuvieco et al., 2018). The MODIS C6 burned area is generated from the MODIS C6 Terra and Aqua 500-m daily surface reflectance products, MOD14A1 and MYD14A1 1-km daily level 3 active fire products, and the MCD12Q1 500-m annual land cover product (Giglio et

al., 2018).

For present-day fire carbon emissions, we used GFED4s (van der Werf et al., 2017; GFED5 fire emissions have not been released), the Global Fire Assimilation System (GFAS1.2; Kaiser et al., 2012), and the Fire Energetics and Emissions Research (FEER-G1.2; Ichoku and Ellison, 2014) as benchmarks. The GFED4s fire emissions since 1997 at 0.25° are constructed using the CASA

biogeochemical model with GFED4s burned area and observed meteorology and vegetation as inputs (van der Werf et al., 2017). The 0.1° daily GFAS1.2 from 2003 to the present is based on observations of fire radiative power (FRP) from the MODIS and the biome-specific conversion factors derived based on GFED3.1 dry matter burned. The 0.5° daily FEER-G1.2 fire emissions since 2003 are derived from MODIS FRP and constrained with the MODIS aerosol optical depth (AOD) product MOD04_L2

(Ichoku and Ellison, 2014). The three fire carbon emissions products represent the range of satellite-derived inventories well (Li et al., 2019; Wiedinmyer et al., 2023). We used the 2003-2014 period of these satellite products as it overlaps with the timeframe of the CMIP6 historical simulations.

To evaluate long-term trends in fires starting from 1850, we used 992 charcoal records from the Reading Paleofire Database (RPD; Harrison et al., 2021). Sedimentary charcoal records reflect changes

in biomass burning, which are primarily influenced by burnt area (Haas et al., 2023), but are also affected by combustion completeness. In the CMIP6 simulations, biomass burning is represented by the variable of fire carbon emissions, and models submitted fire carbon emissions are much more than those submitted burned area. Consequently, this study examined the similarity in trends between these records and simulated fire carbon emissions. The largest number of records in the RPD are from North

America and Europe, but there are enough records for other regions to construct trustworthy regional composites except for Northern Hemisphere Africa (NHAF), Southern Hemisphere Africa (SHAF), and the Middle East (MIDE) (Fig. S1).

**2.3 Simulations and observations of fire drivers**

We downloaded the CMIP6 model outputs of precipitation and sea surface temperature (SST) from

http://esgf-node.llnl.gov/search/cmip6/ (last accessed: March 2023), and CMIP5 precipitation

simulations from http://esgf-node.llnl.gov/search/cmip5/ (last accessed: March 2023).

Observed 0.5° monthly precipitation observations were obtained from the Climatic Research Unit

(CRU TS v.4.04) (Harris et al., 2020). The 1870−2023 1° monthly SST observations were from the

Hadley Centre Sea Ice and Sea Surface Temperature data set (HadISST) (Rayner et al., 2003). Annual

population density data from 1850 to 2014 at 0.5° spatial resolution were taken from HYDEv3.2

(Goldwijk et al., 2017), which were also used to drive the CMIP6 models.

**2.4 Data processing**

For CMIP6 fire simulations, we corrected unit errors and then uniformly adopted % $mon^{-1}$ for the

burned area fraction and kg C $m^{-2}$ $s^{-1}$ for fire carbon emissions. The data were then regridded to a 1°

spatial resolution using bilinear interpolation for coarser-resolution simulations and area-weighted

averaging for finer-resolution simulations and satellite-based products.

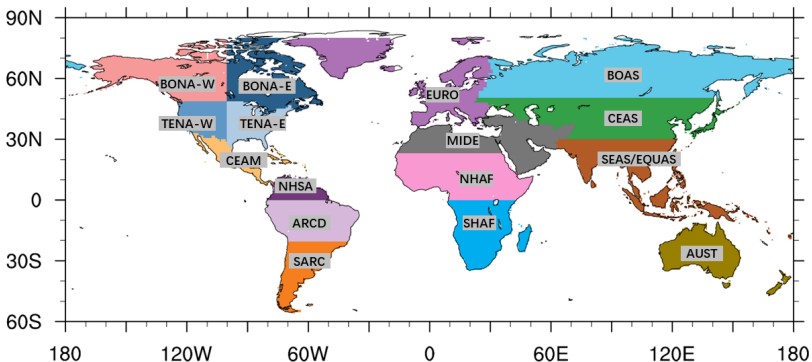

**Fig. 1.** The definition of 16 regions used in this study which combines the GFED regions and RPD

regions. The abbreviations are BONA-W: boreal North America- west; BONA-E: boreal North

America-east; TENA-W: temperate North America- west; TENA-E: temperate North America-east;

CEAM: Central America; NHSA: Northern Hemisphere South America; ARCD: Arc of deforestation;

SARC South of the arc of deforestation; EURO: Europe; BOAS: boreal Asia; MIDE: Middle East;

CEAS: central Asia; NHAF: Northern Hemisphere Africa; SHAF: Southern Hemisphere Africa;

SEAS/EQUAS: Southeast Asia/equatorial Asia; AUST: Australia.





For RPD charcoal records, we constructed composite time series for different regions (Fig. S1) after rescaling the individual records using a minimax transformation, homogenizing the variance using the Box-Cox transformation, and rescaling the transformed values to z-scores (Power et al., 2010). The composite curve for each region was constructed with decadal resolution and a base period from 1750 to 2010. The loess regression with a half-window width of 10 years was used to yield estimates for each decade. Uncertainties (95%) were calculated by bootstrap resampling of the records 1000 times.

For the regional analysis, we divided the global land into 16 regions (Fig. 1). This was done by combining the 14 GFED regions with the 12 RPD regions.

The Niño3.4 index, a widely recognized indicator for El Niño-Southern Oscillation (ENSO), is defined as the SST anomaly averaged over the central-eastern equatorial Pacific (5°S–5°N, 120°–170°W). Based on this definition, we calculated the simulated and observed Niño3.4 index.

## 2.5 Evaluation methods

Our evaluations focus on the total amount, spatial distribution, seasonal cycle, long-term trend, and interannual variability of burned area and fire carbon emissions, as well as the relationship between fires and climatic or socioeconomic factors.

The global and regional totals or averages of a variable were calculated as the area-weighted sum or average across global land areas and specific regions, respectively. The Pearson correlation coefficient between observations and simulations was used to evaluate the skill of spatial and temporal variability patterns, and the Student's t-test was used to assess its significance. For testing the significance of the spatial correlation, the effective degrees of freedom (EDF) were estimated via the widely used method of Bayley and Hammersley (1946) and Clifford et al. (1989), in which autocorrelation in observed and simulated spatial patterns reduces the EDF, thereby raising the thresholds for statistical significance.

We estimated the long-term trend using the ordinary least squares (OLS) method and evaluated its significance using the Mann-Kendall test.

The coefficient of variation (CV, standard deviation divided by the average) was used to quantify the magnitude of interannual variability and seasonality. Given that ENSO is the dominant driver of the interannual variability of pan-tropical fires (Chen et al., 2017), we evaluated simulations of fire interannual variability using correlation between detrended tropical fires and the detrended Niño3.4





index.

When evaluating the relationship between fire and its drivers, we examined how the annual burned area fraction in the tropics and subtropics (35°S to 35°N) varied with annual precipitation, following Prentice et al. (2011) and Kloster and Lasslop (2017), and how global annual burned area

fraction changed with population density as in Li et al. (2018).

### 3. Results

#### 3.1 Global totals

The present-day global burned area estimated by six out of nine CMIP6 models and the ensemble mean

(MME) fall within the range of satellite-based products (430–802 Mha yr$^{-1}$) (Fig. 2a). CMIP6 models perform much better than CMIP5 models (150– 184 Mha yr$^{-1}$, below half the area shown by the benchmarks). The inter-model discrepancy of CMIP6 models is larger than the CMIP5 models, primarily due to the big difference in the three models that incorporate the GLobFIRM fire scheme and its variant (764, 172, and 176 Mha yr$^{-1}$ for CNRM-ESM2-1, EC-Earth3-CC, and EC-Earth3-Veg,

respectively). CNRM-ESM2-1 tuned the parameters of GlobFIRM based on fire occurrence measurements to obtain a more reasonable estimate of the global burned area (Delire et al., 2020).

The global totals of fire carbon emissions estimated by 11 out of 18 CMIP6 models fall within the range of benchmarks, as is the MME (Fig. 1b). The CMIP6 ensemble mean outperforms the CMIP5 ensemble mean, although inter-model differences are larger mainly due to the anomalously low value

of NorCPM1. Overall, EC-Earth3 models in CMIP6 and most CMIP5 models that use the GlobFIRM scheme reasonably simulate the global total of fire carbon emissions, even though the estimated global burned area is less than half of the observed values. This is mainly because GlobFIRM uses higher combustion completeness factors for woody tissues (70–90% for stem and coarse woody debris) than those used in Li (27–35% for stem and 40% for coarse woody debris) and SPITFIRE (0–73% for 100

hr fuel type and 0–41% for 1000 hr fuel type) (Li et al., 2019) and the satellite-based GFED family (20–40% for stem and 40–60% for coarse woody debris) (van der Werf et al., 2017). CNRM-ESM2-1 employs GlobFIRM but adjusts the completeness factors down to obtain a reasonable estimate of fire carbon emissions (Delire et al., 2020).



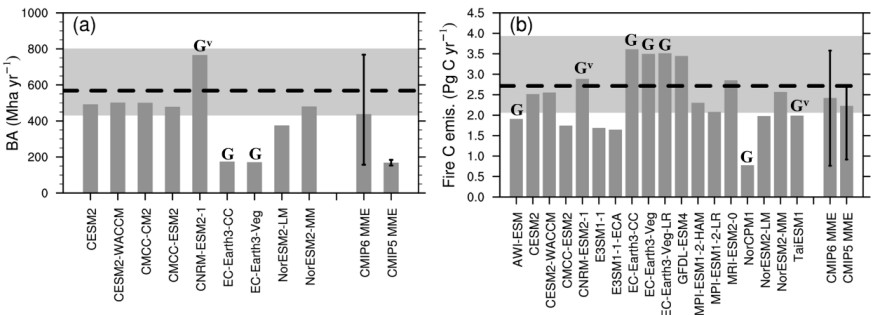

**Fig. 2.** Present-day global totals of (a) burned area and (b) fire carbon emissions for benchmark averages (dashed lines) and model simulations (bars). The shaded areas show the range of the benchmarks and error bars the range of the models. The assessment is made for 2001–2014 for burned area and 2003–2014 for fire carbon emissions. The CMIP multi-model ensembles (MMEs) span 2001–2005 for burned area and 2003–2005 for fire carbon emissions. G and $G^v$ denote models that use GlobFIRM or its variant. Other models used the Li fire scheme in (a) and Li or modified SPITFIRE in (b).

### 3.2 Spatial pattern

All the CMIP6 models capture key features of the observed spatial pattern of present-day burned area fraction, with global spatial correlations between simulations and observations that are statistically significant at the 0.05 level (Fig. 3). CMIP6 models outperform CMIP5 models in simulating spatial patterns, with correlation coefficients increasing from a range of 0.15–0.34 in CMIP5 (Fig. S2) to 0.28–0.70 in CMIP6. CMIP6 models incorporating the Li fire scheme have even higher correlations, ranging from 0.54 to 0.70. Most CMIP6 models successfully reproduce the observed high values in Africa, except for EC-Earth3-CC and EC-Earth3-Veg, which both use GlobFIRM (Fig. 3). However, most CMIP6 models overestimate burned area in the South American savannas (Fig. 3; Table S1). Additionally, models using GlobFIRM overestimate the burned area in the western United States and tropical rainforests, whereas those using the Li scheme typically underestimate it in boreal shrublands possibly because the wet and cold bias during fire season in the ESM climate simulations (Figs. S3c and 4c) leads to the underestimations of fuel flammability.

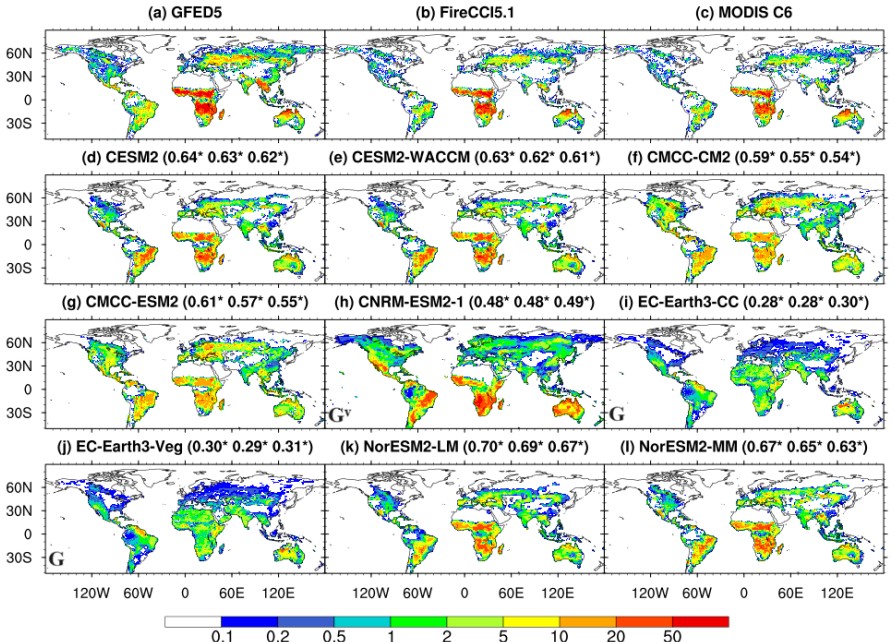

**Fig. 3.** 2001–2014 spatial distribution of annual burned area fraction (% yr⁻¹) for (a–c) benchmarks and (d–l) CMIP6 models. The spatial correlations of simulations with three benchmarks are also given in parentheses. *: correlation significant at the 0.05 level based on the Student-t test with estimated effective degrees of freedom (EDF), which considers the impacts of autocorrelation in observed and simulated spatial patterns. G and Gᵛ denote models using the GlobFIRM fire scheme and its variant, respectively, while the other models use the Li scheme.

The CMIP6 models skillfully reproduce the observed spatial pattern of fire carbon emissions, except for MRI-ESM2 which utilizes GlobFIRM and incorrectly places the areas of high emissions north of 45ºN (Fig. 4). The incorrect simulations of MRI-ESM2 (Fig. 4q) are likely due to a wet bias, and particularly, a large warm bias north of 45ºN outside the fire season (Fig. S5), which contributes to the accumulation of fuel for burning. Compared to CMIP5 models (Fig. S6), CMIP6 models improve the simulations of spatial patterns, but the improvement is not as evident as that for the burned area fraction. The bias in the simulations of fire carbon emissions (Fig. 4, Table S2) is similar to that for burned area fraction simulations. Models incorporating the complex SPITFIRE scheme do not



outperform simpler fire schemes, showing similar overestimations in the western United States and

Arctic tundra as those using GlobFIRM (Fig. 4).

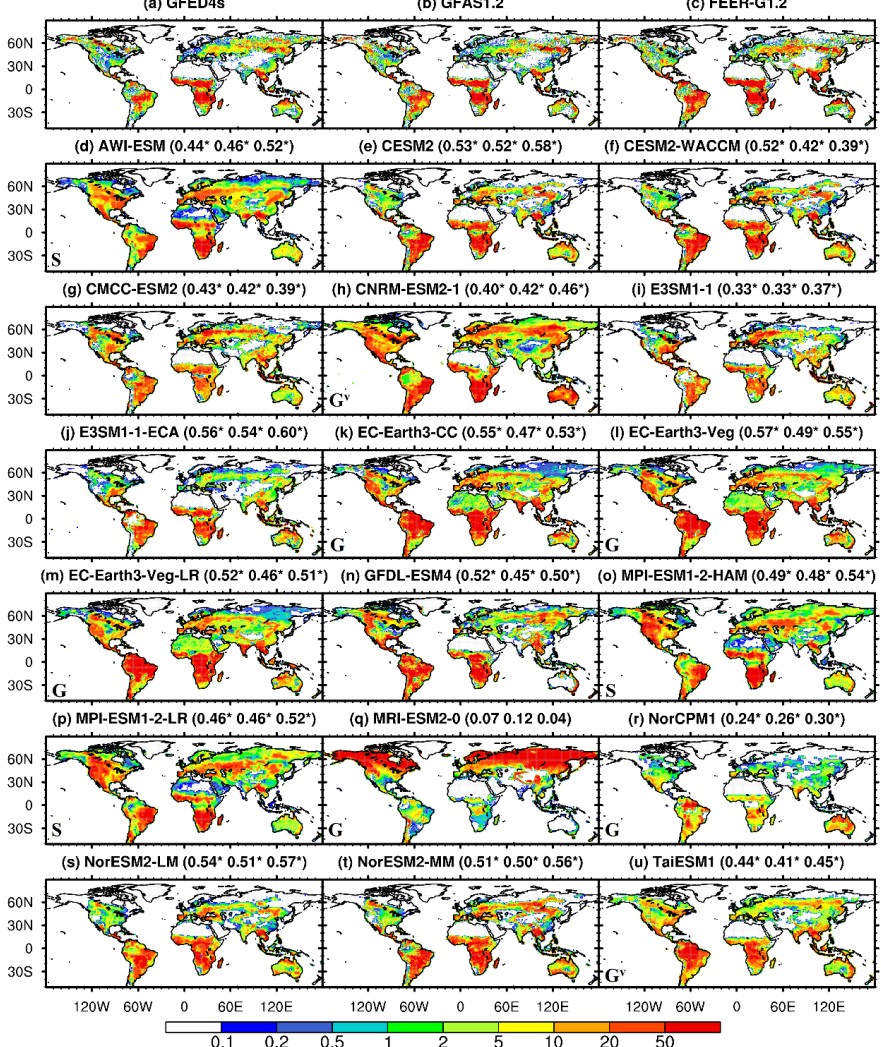

**Fig. 4.** Same as Fig. 3, but for 2003–2014 fire carbon emissions (g C m$^{-2}$ yr$^{-1}$) for (a–c) benchmarks

and (d–u) CMIP6 models. S indicates models using the SPITFIRE fire scheme.

The CMIP6 MME exhibits an improved skill in simulating spatial patterns compared to CMIP5

MME, particularly for burned area fraction (Fig. 5). The global spatial correlation between simulations

and observations for the CMIP6 MME is 0.69, more than twice that for the CMIP5 MME (0.30). The

most notable improvement in the burned area simulations is that CMIP6 models capture the

observational high values in tropical savannas across Africa, South America, and Australia, as well as

the moderate values observed in the boreal forests in Eurasia (Figs. 5a–c). CMIP6 MME outperforms

CMIP5 MME for fire carbon emissions mainly by reducing the underestimation in Asian boreal forests

5    and South Asia as well as the overestimation in western North America and South America (Figs. 5d–

f). However, there are outliers: the EC-Earth models, for example, overestimate both the burned area

(Figs. 3i–j) and fire carbon emissions (Figs. 4k–l) in the arid regions of northern Africa, while MRI-

ESM2 overestimates fire carbon emissions in regions north of 45ºN (Fig. 4q).

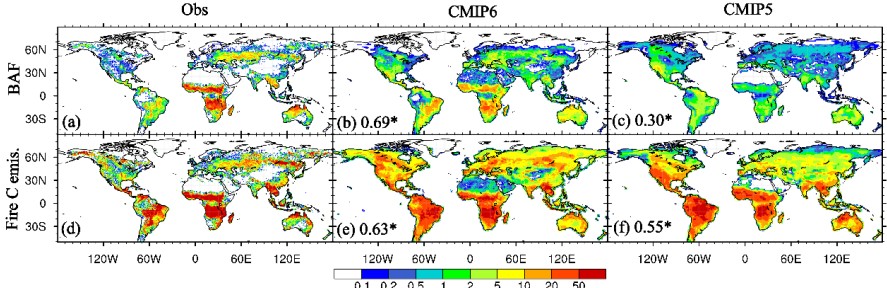

**Fig. 5.** Spatial distribution of (a–c) annual burned area fraction (BAF, % yr $^{-1}$) averaged over 2001–

2005 and (d–f) annual fire carbon emissions (g C m$^{-2}$ yr$^{-1}$) averaged over 2003–2005 for benchmark

average (Obs), CMIP6 MME, and CMIP5 MME. The global spatial correlation between simulation and

observations is also given, with * representing a correlation significance at the 0.05 level.

15    **3.3 Seasonal cycle**

The CMIP6 models capture major features of the burned area seasonality: peak month occurs in the dry

season in the tropics and in the warm season in the extra-tropics (Fig. 6). The CMIP6 models

accurately capture the peak fire month in July-August in NH high-latitudes (Figs. 6a–b) and January in

SH mid-latitudes (Figs. 6i–j). The temporal correlation with observations ranges from 0.87 to 0.95 for

20    the NH high-latitudes and 0.57 to 0.70 for the SH mid-latitudes, all significant at the 0.05 level. On the

contrary, most CMIP5 models fail to capture the seasonal phase of these regions (Figs. 6b and 6j).

In the NH tropics, the peak timing of CMIP6 models and the MME occurs in March, which is later

than the observed peak in December-January (Figs. 6e–f). Despite this, they still outperform the

CMIP5 models, which peak in March-May. In the SH tropics, both CMIP6 and CMIP5 models exhibit



similar timing, peaking one or two months later than observed (Figs. 6g–h). The delays in fire peak

timing for both CMIP5 and CMIP6 models are partly attributed to a simulation bias in precipitation,

where the month with the minimum precipitation in the models occurs one or two months later than

observed (Fig. S7).

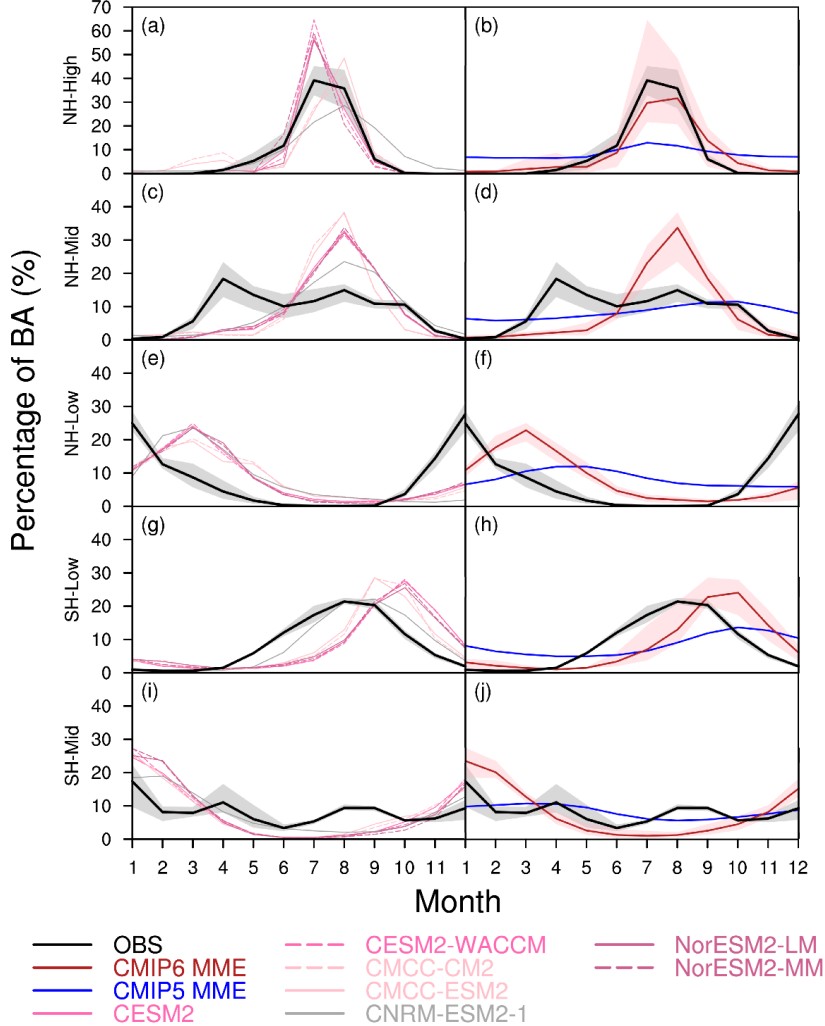

**Fig. 6.** Seasonal cycle of burned area for observations and (left) CMIP6 models averaged over 2001–

2014 or (right) CMIP6 and CMIP5 MMEs averaged over 2001–2005. Shaded areas are a range of

benchmarks or models. EC-Earth3-CC and EC-Earth3-Veg do not model fire seasonal cycles and are

10    thus excluded.





In the NH mid-latitudes, there are two observed fire peaks: one in spring mainly caused by crop fires, and the other in summer caused by fires occurring in natural vegetation areas (Fig. 6c). The CMIP6 models reproduce the summer peak with greater accuracy than the CMIP5 models, which peak

two months later than the observations. However, neither the CMIP6 models nor the CMIP5 models capture the spring peak (Figs. 6c–d).

CMIP6 models reasonably simulate the magnitude of the seasonal variability of burned area, and outperform CMIP5 models, except for the SH mid-latitudes (Fig. 6; Table 2). CMIP5 models severely underestimate the seasonal variation across all regions, simulating an abnormally flat seasonal cycle

that represents only about 30% of the observed variation (Table 2).

**Table 2.** Skill scores of seasonality simulations for CMIP6 and CMIP5 MMEs. The correlation coefficient and coefficient of variation (CV) are used to evaluate the phase and magnitude of seasonal variability, respectively. *correlation significant at the 0.05 level.

|  | NH-high | NH-mid | NH-low | SH-low | SH-mid |
|---|---|---|---|---|---|
|  | correlation coefficient | | | | |
| CMIP6-MME | 0.98* | 0.53* | 0.22 | 0.58* | 0.66* |
| CMIP5-MME | 0.93* | 0.20 | −0.39 | 0.33 | 0.38 |
|  | CV | | | | |
| Obs | 1.69 | 0.74 | 1.17 | 0.96 | 0.43 |
| CMIP6-MME | 1.73 | 1.28 | 0.88 | 1.02 | 0.99 |
| CMIP5-MME | 0.20 | 0.24 | 0.29 | 0.39 | 0.23 |

The performance of CMIP6 models in simulating the seasonal phase of fire carbon emissions is similar to that for burned area (Fig. S8). Models using the Li and SPITFIRE schemes capture the peak in NH high-latitudes (Figs. S8a–b) and the summer peak in NH mid-latitudes (Figs. S8c–d). They exhibit a 1–2 month delay in the fire peak timing in NH and SH low latitudes (Figs. S8e–h). The peak

in the SH mid-latitudes is captured accurately by models using the Li scheme but occurs two months later for SPITFIRE (Figs. S8i–j). In addition, different from the burned area simulations, both CMIP6 and CMIP5 models overall reproduce the observed magnitude of seasonal variability for fire carbon emissions (Fig. S8), although the CMIP5 models underestimate the seasonal variation in NH high-latitudes (Figs. S8a–b) and SH mid-latitudes (Figs. S8i–j).





### 3.4 Trend

In recent decades, satellite-based products have revealed a significant decline in burned area and fire

carbon emissions, but CMIP6 models do not capture this trend (Fig. 7), similar to the CMIP5 models

(Kloster and Lasslop, 2017). Spatially, the observed decline in 2001–2014 burned area is most

5      pronounced in tropical savannas across South America, NH Africa, and Australia, showing significant

trends of –1.9 to –0.4, –3.8 to –2.2, and –2.2 to –1.7 Mha yr$^{-1}$ (range of different benchmarks) for the

three regions, respectively. However, CMIP6 models exhibit trends of –0.6 to 0.8, –0.3 to 0.9, and –1.5

to 0.6 Mha yr$^{-1}$ for these regions. The failure is partly due to the inadequate representation of human

fire suppression efforts in the fire schemes (Andela et al., 2017).

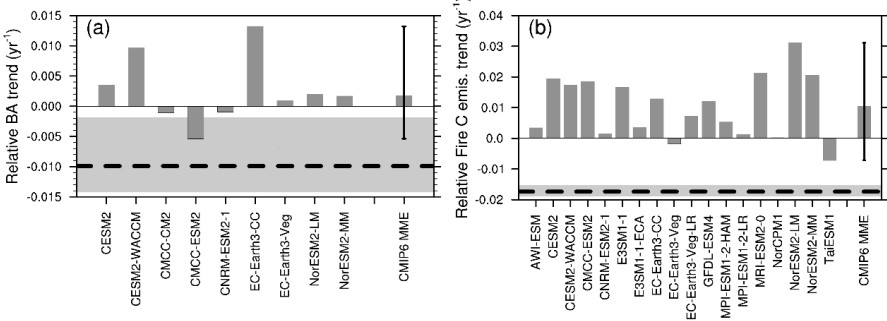

**Fig. 7.** Same as Fig. 2, but for the relative trends in the present day (2001–2014 for the burned area and

2003–2014 for the fire carbon emissions).

15      Looking back to the period starting from 1850, most CMIP6 models and the CMIP6 MME

simulate the change between the present day (1985−2005) and the pre-industrial period (1855−1875)

that have the same sign (either an increase or decrease) as the charcoal-based reconstructions in 11 out

of the 12 regions (Fig. 8). In eastern boreal North America (BONA-E), although the signs differ

between the CMIP6 MME and RPD, both values are very small (Fig. 8b). In contrast, CMIP5 models

20      show trends consistent in sign with the RPD reconstructions in only 4 of the 12 regions (BONA-E,

TENA-E, ARCD, and SEAS/EQAS), indicating poorer performance compared to CMIP6 models (Fig.

8).

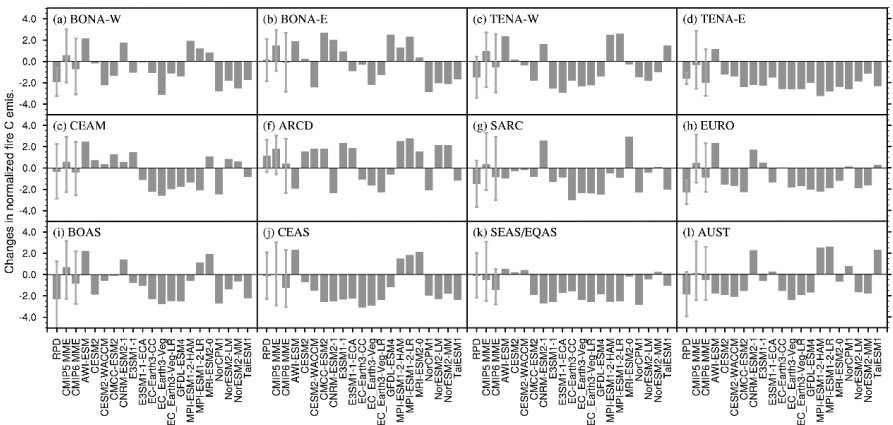

**Fig. 8.** Comparison of normalized fire carbon emissions changes between the present day (1985−2005)
and the pre-industrial period (1855−1875) across different sources: charcoal-based RPD, CMIP6
models, and both CMIP6 and CMIP5 MMEs. The error bars represent uncertainties, calculated as the
average over the present-day and pre-industrial periods for RPD, and as the range across model
simulations for the MMEs.

For time series changes, most CMIP6 models can capture the overall downward trend for 1850–
1990 in western boreal North America (BONA-W), western temperate North America (TENA-W),
Europe (EURO), boreal Asia (BOAS), and Australia (AUST) (Figs. 9a, c, h, i and l) and upward trend
for Arc of deforestation (ARCD) (Fig. 9f) as depicted in RPD. However, simulations and RPD have
different trends in eastern temperate North America (TENA-E) and the southern Arc of Deforestation
in South Africa (SARC) prior to the 1910s (Figs. 9d and g), as well as in temperate North America
(TENA-E and TENA-W), eastern boreal North America (BONA-E), EURO, and BOAS from the
1980s onwards (Figs. 9b–d and h–i). Since the 1980s, CMIP6 simulations have shown an increase in
these regions, whereas the RPD reconstructions show a decline. The long-term (1982–2018) fire
reanalysis product FireCCILT11 supports the rising trend in EURO simulated by the CMIP6 models,
but shows a decrease in BOAS similar to RPD (Otón et al., 2021). Furthermore, the increase in TENA-
W, depicted in most CMIP6 models, is supported by the analyses of remote-sensing-based fire
perimeter datasets (e.g., Abatzoglou and Kolden, 2013; Abatzoglou and Williams, 2016; Williams et
al., 2019), contrary to the RPD.

Most CMIP6 models also show a decline in TENA-E fire emissions before the 1910s, while RPD



and some CMIP5 models suggest an increase (Figs. 9 and S9). From 1850 to 2010, the CMIP6 models

using SPITFIRE simulate increased fire emissions in TENA-W, and the CMIP6 models using

GlobFIRM simulate decreased fire emissions in ARCD, which are not seen in the RPD reconstructions

and CMIP6 models using the Li scheme (Fig. 9c and f).

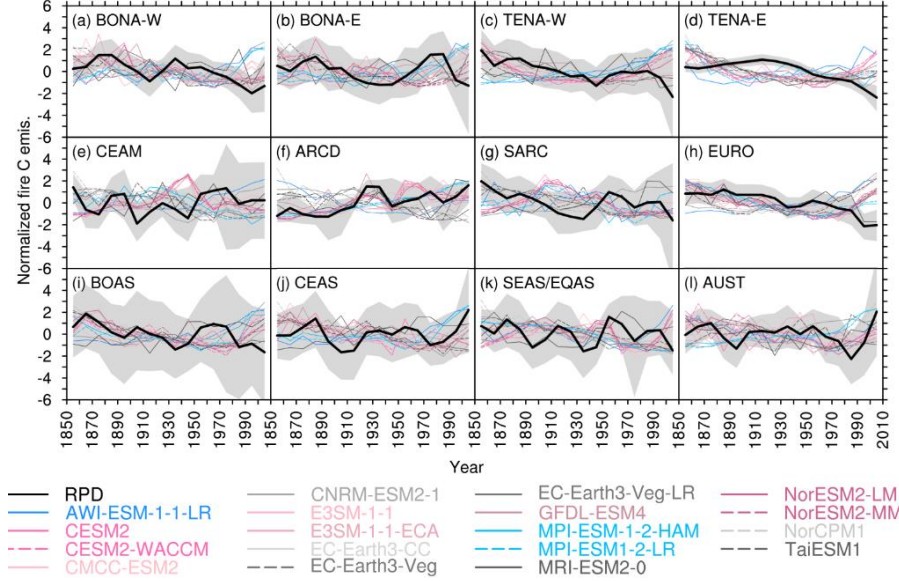

**Fig. 9.** Standardized fire carbon emissions simulated by CMIP6 models and indicated by RPD charcoal

product.

10  **3.5 Interannual variability**

Unlike Dynamic Global Vegetation Models (DGVMs) that are driven by observed climate data (Li et

al., 2019; Hantson et al., 2021), coupled models in CMIP are free-running and driven solely by

anthropogenic forcing. Consequently, they do not aim to synchronize with the actual climate state of

specific years (Taylor et al., 2012; Eyring et al., 2016). Therefore, expecting a one-to-one match

15  between CMIP-simulated and observed fires in any given year is unrealistic. Instead, we evaluate the

magnitude of interannual variability and how fire activity responds to key climate drivers, such as

ENSO, a dominant climate oscillation on an interannual timescale, affecting fires in the tropics (van der

Werf et al., 2006; Prentice et al., 2011; Chen et al., 2017).



CMIP6 models demonstrate large inter-model discrepancies in simulating interannual variation. For burned area, the modifications of GlobFIRM implemented in CNRM-ESM2 (Delire et al., 2020) and the updates to the Li scheme employed by CESM2 and the NorESM family (Li and Lawrence, 2017) weaken interannual variation compared to EC-Earth3 and the CMCC family, respectively (Fig. 10a). For fire carbon emissions, models using the Li scheme overestimate the interannual variation, while those using the SPITFIRE underestimate it (Fig. 10b).

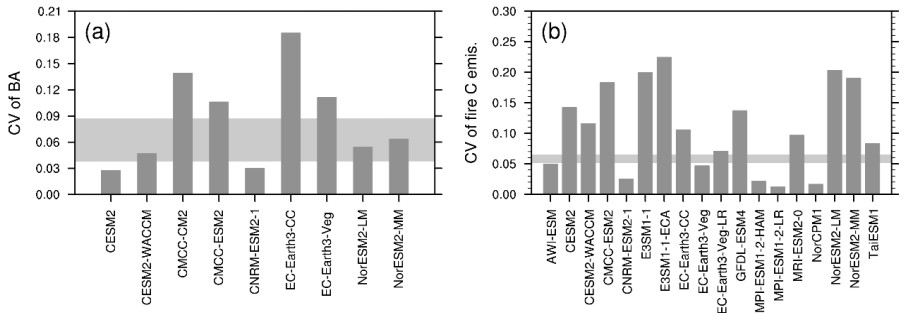

**Fig. 10.** Same as Fig. 7, but for coefficient of variability (CV) of interannual variability.

The warm phase of ENSO (El Niño), characterized by warm SST anomalies in the tropical central-eastern Pacific (quantified by the Niño3.4 index), is typically initiated during the boreal summer and persists through the following spring, reaching its peak in boreal winter. Here, we assess the influence of winter (DJF) El Niño on the interannual variability of annual tropical fires averaged from the preceding June to the following May. In observations, El Niño-induced anomalies in the Walker circulation along the equator lead to decreased precipitation, increased fuel flammability, and enhanced burning and fire carbon emissions in equatorial South America and Southeast Asia, whereas they produce the opposite effect in eastern Africa (Fig. S10). In general, the CMIP6 models successfully capture the response of fire carbon emissions in the three regions to El Niño, except for the MPI models using the SPITFIRE scheme (Fig. 11).



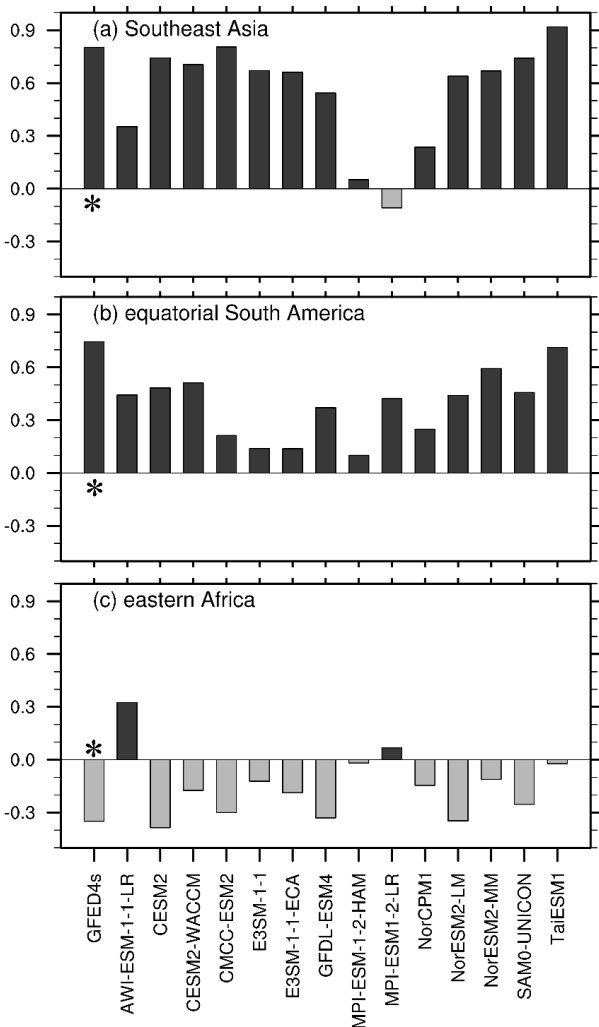

**Fig. 11.** Correlation coefficient between DJF Niño3.4 index and tropical fire carbon emissions averaged from the preceding June to the following May for GFED4s (1997–2019) and CMIP6 models (1850–2014). *: benchmark GFED4s. EC-Earth3 models, which only provide annual total fire emissions at the end of the year, are excluded.

## 3.5 Relationship between fires and climatic and socioeconomic factors

Observations indicate a distinct unimodal relationship between burned area and mean annual precipitation in the tropics and subtropics (Fig. 12a): burned area increases as precipitation increases





due to increased fuel load, peaking at annual precipitation of 1100 mm yr⁻¹, and then declines as

precipitation increases due to decreased fuel flammability (van der Werf et al., 2008; Archibald et al.,

2009). CMIP6 models using the Li scheme reproduce the unimodal relationship, whereas models using

GlobFIRM peak at annual precipitation of 100–500 mm yr⁻¹ (Figs. 12b–j). However, all the CMIP6

models show weaker variability in magnitude than observations, indicating that the sensitivity of fires

to humid conditions is underestimated.

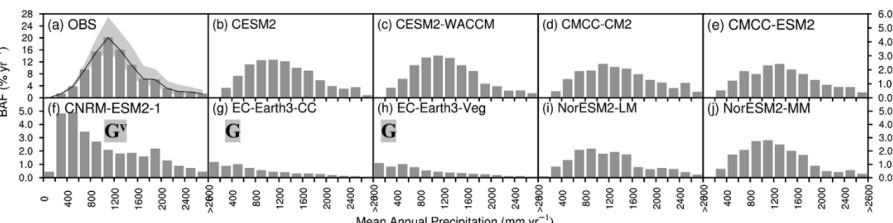

**Fig. 12**. Burned area fraction between 35°N and 35°S in 200 mm yr⁻¹ bin of mean annual precipitation

for (a) benchmarks and (b–j) CMIP6 model simulations. G and Gᵛ denote models with the GlobFIRM

and its variant.

CMIP6 models using the Li scheme outperform all CMIP5 models in reproducing the relationship

between fires and wet-dry conditions. All CMIP5 models peak at lower annual precipitation values than

the observations (Fig. S11). Specifically, CESM1-BGC and CCSM4 have a maximum for mean annual

precipitations of around 700–900 mm yr⁻¹, while MPI models exhibit a peak at around 400 mm yr⁻¹.

Similar to CMIP6 models, all CMIP5 models underestimate the sensitivity of tropical and subtropical

fires to humidity, but to a greater extent (Fig. S11).

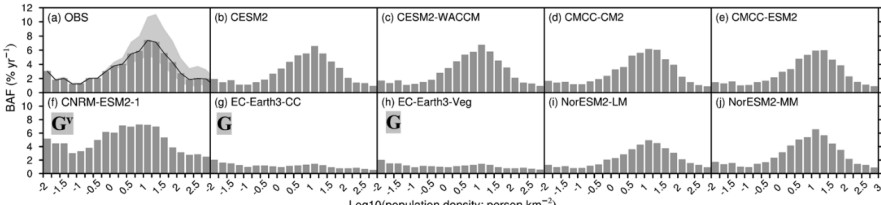

**Fig. 13.** Burned area fraction changes with increasing population density for (a) benchmarks and (b–j)

CMIP6 model simulations.



The observed burned area fraction rises with increasing population density, mainly due to increased human ignitions, peaking at 10–18 person km$^{-2}$, and then falls due to increased human suppression (Fig. 13a) (Pechony and Shindell, 2009; Bistinas et al., 2014; Haas et al., 2022). CMIP6

models using the Li scheme and CNRM-ESM2-1 using the GlobFIRM variant reproduce the observed relationship well, while CMIP6 models using the GlobFIRM and all CMIP5 models fail due to a lack of representation of human influence on fire occurrence and spread (Figs. 13b–j, Fig. S12).

### 4. Conclusions and discussion

**4.1 Summary**

This study provides the first comprehensive evaluation of global fire simulations in CMIP6 ESMs and documents considerable improvements compared to CMIP5 models. Our main findings can be summarized as follows:

- Global totals: Most CMIP6 models, along with the multi-model ensemble mean, estimate global
totals of burned area and fire carbon emissions within the range of satellite-based observations. CMIP6 addresses the major issue identified in the CMIP5 models that simulate a global burned area of less than half of the observed. The increased inter-model range in CMIP6 is due to the inclusion of models using the GlobFIRM fire scheme.

- Spatial pattern: CMIP6 models and the ensemble mean skillfully simulate the spatial patterns of
burned area and fire carbon emissions. Models using the GlobFIRM have around half the skill (measured as spatial correlations) in simulating burned area compared to those using the Li scheme. Models that use the complex SPITFIRE fire scheme do not outperform the other models. Notably, CMIP6 models capture the high burned area fraction observed in Africa, whereas all CMIP5 models fail to reproduce this feature. The global correlation between the CMIP6 simulated
burned area and observations is twice that of the CMIP5 models. Simulations of fire carbon emissions have been improved as well, albeit to a lesser degree.

- Seasonal cycle: CMIP6 models and the ensemble mean capture the major features of the fire



seasonal phase (timing), but fail to reproduce the spring peak at NH mid-latitudes. They also

simulated fire peak timing around two months later than the observed in the tropics, partly due to

the bias in the simulated precipitation. Overall, CMIP6 models outperform CMIP5 models in

replicating the timing.  Importantly, CMIP6 addresses the major issue identified in CMIP5 models,

which simulated burned area seasonal variation (quantified using CV) at only about 30% of the

observed level.

- Long-term trend: CMIP6 models still fail to reproduce the observed significant decline in burned

   area and fire carbon emissions over the past two to three decades, largely due to an

   underestimation of anthropogenic influences that suppress fires. For the period 1850–2010,

simulated regional changes in fire carbon emissions align with the RPD charcoal-based

   reconstructions, except for Southern South America and eastern temperate North America before

   the 1910s and temperate and eastern boreal North America, Europe, and boreal Asia since the

   1980s. CMIP6 simulations are generally closer to RPD than CMIP5 simulations.

- Interannual variability: CMIP6 models can capture the response of interannual variability of

tropical fires to ENSO, except for models using the SPITFIRE fire scheme. However, there are

   large inter-model differences in simulating the magnitude of interannual variation.

- Relationship of fires with precipitation and population density: CMIP6 models capture the

   unimodal relationship between burned area and precipitation in the tropics and subtropics and

   between the global burned area and population density, except for models using the GlobFIRM fire

scheme. CMIP6 models outperform CMIP5 models, but all CMIP6 and CMIP5 models

   consistently underestimate fire sensitivity to precipitation.

**4.2 Reasons for improved fire simulations in CMIP6**

The improved fire simulations from CMIP5 to CMIP6 are mainly attributed to the development of fire

schemes. The most-used fire scheme has evolved to the Li scheme in CMIP6 from the GlobFIRM in

CMIP5. Li et al. (2012, 2013) assessed the two schemes in CLM4 land model offline simulations using

the same inputs (observed climate, lightning frequency, $CO_2$ concentration, land use and land cover

change, socioeconomic conditions) and experimental design. The results indicated that the Li scheme





not only aligned more closely with the observed global burned area, with estimates twice that of GlobFIRM, but also doubled the simulated skill in spatial pattern, notably capturing the high burn area fraction in Africa. Li et al. (2019), Hantson et al. (2020), and Wang et al. (2022) evaluated fire simulations of the DGVMs participating in FireMIP, which used the same protocol and input data, and

confirmed the superiority of the Li scheme.

The Li fire scheme outperforms GlobFIRM primarily due to its superior core equation for calculating burned area and its calibration of parameters and functions based on observations. The Li fire scheme calculates the time-step burned area as the product of fire counts and average fire spread area per fire, in which all variables have observations, allowing for parameter calibration (Li et al.,

2012, 2013). Such calibration has indeed been performed on the parameters, as documented by Li et al. (2021, 2013). GlobFIRM, on the other hand, calculates the annual burned area fraction as a nonlinear function of fire season length, where the fire season length is calculated by summing fire occurrence probability over a year (Thonicke et al., 2001). Since the probability cannot exceed 1, the burned area will be underestimated in grid cells where multiple fires occur in a time step. Furthermore, fire

occurrence probability has no observations, so fire occurrence parameters cannot be calibrated. In addition, because the annual burned area is simulated, models using the GloFIRM do not simulate fire seasonality. To address this, some models modified GlobFIRM to run at sub-daily to monthly time steps by using the weighted difference of running annual mean burned areas between the current and previous time steps (Krinner et al., 2005; Kloster et al., 2010; Kloster et al., 2017). However, this

modification still results in a significant underestimate of seasonal variation of burned area as shown in Fig. 6. Besides the two primary reasons, the incorporation of human influence on fires, even though partially, enhanced the simulations of the global spatial pattern of burned area.

Improvements and changes in climate simulations in CMIP6 also contribute to improved fire simulations compared to CMIP5 in some regions. The CMIP6 models reduce the wet bias in NH Africa

during the fire season (Fig. S3), resulting in higher fuel flammability and increased burned area. CMIP6 models also simulate a warmer climate in the Arctic boreal zone year-round, displaying a larger warm bias or a shift from cold bias in CMIP5 to warm bias (Fig. S4). The warm bias would lead to increased vegetation growth and hence an increase in fuel availability and an increase in fuel flammability during the fire season through increased drying. Besides, CMIP6 models that incorporate

the Li fire scheme simulate biomass and leaf area index (LAI) more reasonably than their model



versions in CMIP5 (Danabasoglu et al., 2020; Seland et al., 2022), which contributes to more accurate estimates of fuel availability.

**4.3 Implications for future fire model development**

Our evaluation results indicate four critical issues in current fire models, suggesting directions for

future model development.

First, CMIP6 models fail to reproduce the observed present-day significant decline in burned area and fire carbon emissions. The observed decline is largely attributed to increased human suppression (Andela et al., 2017). Archibald (2016) and Andela et al. (2017) found that the increase in land fragmentation from cropland and pasture expansion decreases fuel continuity, resulting in less burned

area and lower fire emissions. Nevertheless, no fire scheme in CMIP6 ESMs parameterizes this. Furthermore, although CMIP6 ESMs using the Li scheme include parameterization on how economic development, measured by GDP per capita, enhances fire suppression, this has no impact on the CMIP6 simulations due to the GDP per capita forcing data they used being fixed at year 2000 levels. Therefore, considering the influence of landscape fragmentation on fires and using time-varying GDP

per capita as forcing data could be a promising direction for future model development.

Second, CMIP6 models still underestimate the global burned area, though they are much better than CMIP5 models. The underestimation is partly because the models do not simulate multiday fires, which would allow for larger fires and thus increase the burnt area. Another reason for the underestimation may be that these models calibrated their parameters and functions using older remote-

sensed products. For example, the Li scheme used GFED3 (Giglio et al., 2010) for calibration (Li et al., 2012, 2013), which reports a global total burned area of around half of that indicated in the latest GFED5 product (Chen et al., 2023). Revisiting model calibrations and considering multiday fires would be helpful to improve model performance.

Third, the CMIP6 models fail to reproduce the observed spring peak in fires at NH mid-latitudes.

The peak is mainly attributed to fires that occurred over croplands. While CMIP6 models using the Li scheme do simulate crop fires (Table 1), they underestimate them for two main reasons: (1) the calibration of crop fire parameters in the Li scheme was based on GFED3 (Li et al., 2013) in which crop fires, generally classified as small fires, are much underestimated (Chen et al., 2023); (2) ESMs using the Li scheme and crop growth model (e.g., CESM family and NorESM family) assume no crop





fires in managed croplands. The CMIP6 models using other fire schemes either assume that no fires occur in croplands or treat them as fires occurring in natural vegetation, leading to an underestimation or incorrect timing of crop fires. Recently, Millington et al. (2022) have deepened our understanding of fire use in croplands, detailing the varied purposes for burning which influence the timing of burns and how environmental conditions affect these practices. Additionally, Hall et al. (2024) have developed a global cropland-focused burned area product. Incorporating the information in fire models would be a helpful step to improve model performance.

Finally, all CMIP6 models underestimate fire sensitivity to precipitation, either by increasing fuel loads as precipitation increases in more arid climates or by reducing flammability in more humid climates. This suggests the need to re-examine the parameterizations of fuel build up and to improve the estimation of fuel wetness.

### 4.4 Implications for developing a reliable future fire projection product

A reliable fire projection product is crucial for knowing how fire regimes may change in the future (Pechony and Shindell, 2020; Kloster et al., 2017; Li et al., 2021; Wu et al., 2021; Yu et al., 2022). It not only aids in guiding fire management, but is also necessary for quantifying the influence of future fires on the carbon, water, and energy cycles, climate, and human well-being (Ward et al., 2012; Jiang et al., 2016; Xie et al., 2022; Li et al., 2022; Lou et al., 2023; Park et al., 2024). Despite the clear need, such a product is currently lacking. The future fire emissions forcing for CMIP6, for example, are derived from integrated assessment models (IAMs), which lack the spatial variability within a broad vegetation category (e.g., forest, grassland) across a country as well as the interannual variability (Feng et al., 2020). These data are not based on mechanistic models and cannot realistically represent future fire dynamics.

The CMIP6 fire simulations, based on mechanistic fire models, represent state-of-the-art multi-model source data for generating global projections of future fires. Our evaluation provides valuable insights into how to use them to produce a reliable fire protection product. Clearly, including models that perform poorly, either with respect to burned area (e.g., EC-Earth3-CC, EC-Earth3-Veg) or with respect to emissions (e.g., MRI-ESM2-0, NorCPM1) will downgrade the quality of the multi-model projection. Correcting biases in multi-year averages (val Marle et al., 2017; Lou et al., 2023), the relationship of fires with socioeconomic factors (e.g., population density, GDP per capita, road



density), climatic variables (Xie et al., 2022; Yu et al., 2022), and land cover change (Wang et al., 2023)
would also improve the reliability of the projections.  Finally, instead of relying on multi-model mean
or median values (van Marle et al., 2017; Lou et al., 2023), it is desirable to use a weighted average
approach in which weights are assigned based on model performance when constructing multi-model

ensembles.

*Code and data availability.* CMIP6 and CMIP5 outputs can be accessed through the Earth System Grid
Federation (ESGF) at http://esgf-node.llnl.gov/search/cmip6/ and http://esgf-
node.llnl.gov/search/cmip5/, respectively. For the evaluation, we utilized functions from the NCL

(NCAR Command Language) at https://www.ncl.ucar.edu/Document/Functions/. The post-processing
scripts are available at https://zenodo.org/records/11185326.

*Author Contributions.* FL conceived the research ideas and wrote the manuscript draft. FL, XS, and
ZDL conducted the evaluation analysis. XD downloaded the model outputs from the CMIP website, SH

conducted the pre-processing of charcoal-based historical reconstruction, and FL collected other data.
All coauthors reviewed and edited the manuscript.

*Competing interests.* One of the co-authors, Sam S. Rabin, is a member of the editorial board of GMD.

*Acknowledgements.* This study is co-supported by the National Key Research and Development Program
of China (2022YFE0106500), the Research Council of Norway project (328922), Guangdong Major
Project of Basic and Applied Basic Research (2021B0301030007), and the National Key Scientific and
Technological Infrastructure project "Earth System Science Numerical Simulator Facility" (EarthLab).
This work is a contribution to the LEMONTREE (Land Ecosystem Models based On New Theory,

obseRvations and ExperimEnts) project, funded through the generosity of Eric and Wendy Schmidt by
recommendation of the Schmidt Futures program (SPH, ICP) and to the HORIZON-MSCA FIRE-
ADAPT (The Role of Integrated Fire Management on Climate Change Adaptation for Ecosystem
Services in Tropical and Subtropical Regions) program (SPH). LRL is supported by the Office of Science,
U.S. Department of Energy Biological and Environmental Research through the Energy Exascale Earth

System Model (E3SM) project supported by the Earth System Model Development (ESMD) program



area and the HyperFACETS project supported by the Regional and Global Model Analysis (RGMA) program area. PNNL is operated for the U.S. DOE under contract DE-AC06-76RL01830. VM and RS are supported by the European Union's Horizon 2020 Research and Innovation programme as part of the ESM2025 project (101003536). We appreciate the editor, Dr. Tatiana Egorova, for her valuable suggestions and efforts in facilitating the review process. We also acknowledge the World Climate Research Programme, which, through its Working Group on Coupled Modelling, coordinated and promoted CMIP5 and CMIP6. We thank the modeling groups for producing and making available their model output, the Earth System Grid Federation (ESGF) for archiving the data and providing access, and the multiple funding agencies that support CMIP6 and ESGF.

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
