# Peer review of "Evaluation of global fire simulations in CMIP6 Earth system models"

_Geoscientific Model Development, 2024_

## Author Response (AR1)

We deeply appreciate your encouraging evaluation of our work and manuscript, as well as your constructive comments and suggestions. Our point-by-point response is provided below, and the manuscript has been revised accordingly to address your valuable comments and suggestions.

[Reviewer 1]

Major concern:

Each model in CMIP5/6 has diverse patterns in physical climatic factors that can modulate fire activities such as temperature, precipitation, relative humidity, and so on, because historical scenario results are from coupled climate simulations with online atmosphere, land, and ocean models. In CMIP6, a "land-hist" scenario is available, which is driven by Global Soil Wetness Project phase three (GSWP3), forcing data and offline land surface results. Therefore, comparing fire activity data (e.g., burnt area and C emission by fire) in a "land-hist" scenario would be a better comparison with the same given climatic forcing in different fire schemes and land surface models. The current result might be influenced by climatic simulation results, and we cannot separate pure impact from the fire scheme's contribution to different fire activity simulation results. Please consider employing a "land-hist" scenario for your evaluation of fire activity simulation performance.

Detailed "land-hist" scenario information below:

https://view.es-doc.org/index.html?renderMethod=id&project=cmip6&id=a77b98df-92df-453f-a506-735ba743ca74&version=1

**Reply:** Thank you for the insightful suggestion regarding the use of the 'land-hist' scenario for evaluating fire simulations in CMIP6. We agree that the simulations of physical climate factors can influence fire simulations in coupled models, but we chose not to evaluate the 'land-hist' scenario for several reasons:

(1) Previous Offline Evaluations: The Fire Model Intercomparison Project (FireMIP) has already conducted offline evaluations of land surface models/DGVMs with different fire schemes, using consistent protocols and forcing (e.g., observed climate forcing CRUNCEP). FireMIP included all fire models used in CMIP6. Although the evaluation results for burned area and carbon emissions were published in separate papers, we already know the large difference in fire simulations from different fire models, e.g. the GlobFIRM and Li which are dominant fire models used in CMIP5 and CMIP6 models.

(2) Key Findings Mostly Independent of Climate Biases: As discussed and analyzed in Sec. 4.2 (the difference between GlobFIRM and Li fire scheme and climate simulation bias during and out of fire seasons are evaluated), our main conclusions about CMIP6's improvements over CMIP5 and remaining deficiencies are primarily attributable to differences in fire models rather than climate simulation biases. These include:

a) Addressing critical issues from CMIP5 (e.g., global burned area Beijing less than half of observations, failure to reproduce high burned area fraction in Africa, and weak fire seasonal variability).

b) Persistent challenges in CMIP6 (e.g., failure to reproduce the observed decline in global burned area and fire carbon emissions over the past two decades due to underestimation of human fire suppression, and underestimation of the spring fire peak in NH mid-latitudes due to underestimation of crop fires).

These conclusions are supported by FireMIP evaluations and previous studies using land surface models that differ only in fire schemes.

(3) Limited Data Availability: We attempted to download 'land-hist' fire simulations but found that only 5 models submitted fire simulation data, significantly fewer than the 19 CMIP6 coupled models.

(4) Focus on Coupled Model Evaluation: Our study aims to evaluate CMIP6 coupled models to understand potential biases in historical and future fire simulations, and to support development of post-processing methodologies for generating reliable fire projection products using CMIP6 multi-model outputs.

(5) Future Plans: We have registered FireMIP as a MIP for CMIP7, where fire models will be advanced versions of those in FireMIP. In CMIP7 fire simulation evaluations, we plan to consider both coupled models and offline land models. The evaluation of offline land models will be more pertinent in the CMIP7 context. We appreciate the reviewer's valuable perspective, which will inform our future research directions, particularly in the context of CMIP7 evaluations.

Minor comments:
L9-10 on P10: ensemble mean (MME) -> multi-model ensemble (MME)
**Reply**: Changed as your suggestion.

L13 on P10: big -> significant or large
**Reply**: "big" has been changed to "large".

L5 on P11: remove (MMEs)
**Reply**: We have removed MMEs according to your suggestion.

L12 on P11: at the 0.05 level -> at the 95% confidence level (needed to modify throughout the whole manuscript)
**Reply**: We appreciate the reviewer's suggestion to change "significant at the 0.05 level" to "significant at the 95% confidence level". However, we prefer to maintain the use of "0.05 level" as it directly specifies the significance level used in our hypothesis testing (Student's t-test for correlation) and is statistically more precise in the context of Probability Theory and Statistics.

[Reviewer 2]
**Introduction**:
1. The introduction effectively sets the context by explaining the role of fire in the Earth system and the advancements in fire modeling from CMIP5 to CMIP6. Consider elaborating more on the specific limitations of previous fire models to strengthen the study's rationale.

**Reply**: In Paragraph 3, we have added specific limitations of GlobFIRM which are the most used fire scheme in CMIP5 as "The most used fire scheme in CMIP5 models was GlobFIRM (Thonicke et al., 2001), which has several limitations that explain the shortcomings in CMIP5 coupled model fire simulations. GlobFIRM calculates annual burned area fraction as a nonlinear function of fire season length, which is determined by summing fire occurrence probability over a year. This approach leads to underestimation in grid cells where multiple fires occur in a single time step because the probability cannot exceed 1. Additionally, the lack of observational data for fire occurrence probability makes it impossible to calibrate fire occurrence parameters. GlobFIRM's annual burned area simulation cannot capture fire seasonality. While some models modified GlobFIRM to operate at sub-daily to monthly time steps by using weighted differences of running annual mean burned areas, Kloster and Lasslop (2017) evaluation showed that this modification did not result in skillful simulations of burned area seasonality."

2. The following two sentences contradict each other:
Page 2, Line 7: "However, the CMIP6 models still fail to reproduce the decline in global burned area and fire carbon emissions observed over the past two decades, mainly attributed to an underestimation of anthropogenic fire suppression,"
Page 2, Line 25: "Despite a reduction in the global burned area over the past two decades, emissions from forest fires and the occurrence of extreme fires have increased (Andela et al., 2017; Zheng et al., 2021)."
Are you stating a decline in fire carbon emissions along with burned area or an increase in fire carbon emissions in spite of a burned area decline?
**Reply:** The two sentences are not contradictory, but we appreciate the opportunity to clarify. **Global** burned area and fire carbon emissions decrease (Andela et al., 2017), but fire carbon emissions for **forest** fires increase (Zheng et al., 2021). This discrepancy is due to the diverse global land cover types, including grasslands, croplands, and savannas, in addition to forests. The decrease in global burned area is largest in savannas (Andela et al., 2017).

**Methods**
The methodology is robust, with detailed descriptions of data sources, fire schemes, and evaluation metrics. The use of multiple satellite-based products and charcoal records for validation is commendable. Including a flowchart summarizing the methodological framework would enhance clarity. **Page 5, Line 9:** "The SPITFIRE scheme is the most complex since it uses the Rothermel model to calculate the fire spread rate in the downwind direction," - Li et al also uses the Rothermel scheme, so mentioning it as a distinction of SPITFIRE is confusing.
**Reply**: We appreciate the suggestion to include a flowchart. However, we believe that Sections 2.4 and 2.5 already provide a clear, step-by-step description of our data processing and evaluation methods. Given that these methods are straightforward, commonly used in evaluation studies, and presented sequentially in the text, we feel that a flowchart may not add significant value to the clarity of our methodology.

Li et al. uses a simple empirical function $u_p = u_{max}C_m g(W)$ to calculate the fire spread rate in the downwind direction $u_p$ (Eq. 14 in Li et al., 2012), where $u_{max}$ is a PFT-dependent parameter, and the latter two items represent the dependence of $u_p$ on fuel wetness and wind speed, respectively. Li doesn't use the complex Rothermel scheme to calculate $u_p$. To avoid confusion, we have added "On the other hand, Li's scheme employs a simple empirical function in which the fire spread rate in the downwind direction is determined by fuel wetness and wind speed, and GlobFIRM does not calculate the fire spread rate.".

**Results**:
The results section is thorough, covering global totals, spatial patterns, seasonal cycles, trends, interannual variability, and relationships with climatic and socioeconomic factors. The documented improvements in CMIP6 models over CMIP5 are well-supported by clear figures and tables. A more detailed discussion on regional discrepancies and their potential causes would be beneficial.
**Reply**: We appreciate your suggestion to provide more detailed discussion on regional discrepancies. We would like to point out that regional analyses are indeed included throughout our paper. Sec. 3.2 and Tables S1 and S2 provided information on regional totals. Sec. 3.4 and Fig. S11 discussed regional trends, both for recent decades and since 1850. Sec. 3.3 focused on seasonality, which inherently addressed regional differences due to the varying seasonal patterns across different regions.

**Page 10, Line 10**: add the word "lower" - "fall within the **lower** range of satellite-based products"
**Reply**: We have not added "lower" because we mean simulations are within the range of satellite-based products (i.e., from 430 to 802 Mha/yr).

Please address the reason why some fire models simulate burned area and fire carbon emissions over the Sahara (Figures 3-5).
**Reply**: The simulation of burned area and fire carbon emissions over the Sahara in EC-Earth3 family models may be partly due to overestimation of fuel load. EC-Earth3 family models use LPJ-GUESS (a DGVM) to simulate vegetation structure and distribution. EC-Earth3 overestimates leaf area index (LAI) (Song et al., 2021), which can lead to an overestimation of carbon in terrestrial ecosystems and thus fuel load. We cannot confirm if the fire overestimation is due to incorrect vegetation distribution simulation, as EC-Earth3 family did not submit vegetation distribution simulations to CMIP6. Observations suggest bare soil should dominate this region. We don't think the overestimation is due to the GlobFIRM fire model itself, as it considers fuel load limitation and assumes only 0.1%/yr annual burned area fraction when fuel load is below 200 g C/m2.
We have added the following text: "The EC-Earth models, for instance, overestimate both burned area (Figs. 3i–j) and fire carbon emissions (Figs. 4k–m) in the Sahara region, likely due to an overestimation of fuel load (Song et al., 2021). As a result, the

CMIP6 MMEs show some burned area and fire carbon emissions over the Sahara (Figs. 5b and e) due to these EC-Earth3 simulations.".

In section 3.3 and perhaps throughout the Results section it would be helpful to spell out whether biases in results are due to biases in drivers or in the components existing/missing from the fire scheme, potentially with a table stating the dominating factors.
**Reply:** Thanks for your comments. We have discussed the bias sources in fire simulations are mainly due to biases in drivers or in the fire scheme and the reasons in Secs. 4.2 and 4.3 for different aspects.

**Conclusions and Discussion**:
The summary effectively summarizes the key findings and their implications. The discussion appropriately contextualizes the findings within the broader literature and identifies critical issues in current fire models. The suggestions for future model development are insightful and practical.
**Reply:** We greatly appreciate the reviewer's positive comments on our summary, discussion, and suggestions for future work. We're glad that our efforts to effectively present and contextualize our findings have been recognized.

**Technical Corrections:**
**Page 2, Line 3**: Replace "the simulated global burned area less than half of the observations" with "the simulated global burned area **is** less than half of the observations"
**Reply**: Thank you for your suggestion. We have added 'being' rather than 'is' because the sentence is part of a list structure, and using 'being' maintains parallel construction with the other two items (i.e., "the failure to reproduce the high burned area fraction observed in Africa" and "the weak fire seasonal variability") in the list.

**Page 10, Line 15**: Replace "are a range" with "show a range".
**Reply:** We have changed to "show a range" according to your suggestion.

**[Reviewer 3]**
This paper presents a comprehensive assessment of fire simulations in 19 Earth System Models (ESMs) from CMIP6, representing a significant and timely contribution to the field. The authors systematically evaluate CMIP6 models' performance in simulating global and regional fire characteristics by comparing them with multiple satellite-based products and charcoal-based historical reconstructions. The results demonstrate substantial improvements in CMIP6 models across several aspects, including the simulation of global burned area, reproduction of high burned area fractions in Africa, and capture of fire seasonal variability. The authors also identify persistent issues, such as the failure to reproduce the observed decline in global burned area and fire carbon emissions over the past two decades, and the

underestimation of spring fire peaks in Northern Hemisphere mid-latitudes. In conclusion, this is a high-quality research paper that significantly contributes to understanding and improving fire simulations in Earth System Models. After minor revisions, this paper will provide valuable insights for researchers and model developers in related fields.

**Reply**: We are deeply grateful for your thorough review and positive assessment of our manuscript. Your recognition of the comprehensive nature of our study and its significant contribution to understanding and improving fire simulations in Earth System Models is greatly appreciated, and we are encouraged by your thoughtful feedback.

Detailed comments:

Page 3, Line 10: What are the potential reasons for CMIP5 models underestimating the simulated burned area?

**Reply:** Thank you for this important question. The underestimation of burned area in CMIP5 models can be primarily attributed to limitations in the most used fire scheme, GlobFIRM. Specifically: GlobFIRM calculates the annual burned area fraction as a nonlinear function of fire season length, where the fire season length is calculated by summing fire occurrence probability over a year (Thonicke et al., 2001). Since the probability cannot exceed 1, the burned area will be underestimated in grid cells where multiple fires occur in a time step. Furthermore, fire occurrence probability has no observations, so fire occurrence parameters in GlobFIRM cannot be calibrated. We have discussed these reasons in detail in Section 4.2, where we explain why CMIP6 fire simulations show improvements over CMIP5. Additionally, following Reviewer 2's suggestion, we have included this information in the third paragraph of the Introduction to provide context early in the paper.

Page 9, Line 10: Please specify the data source for the observed sea surface temperatures.

**Reply:** The observed sea surface temperatures are from Hadley Centre Sea Ice and Sea Surface Temperature data set (HadISST). The information was listed in Sec. 2.3 (Simulations and observations of fire drivers).

Page 11, Line 16: How well do EC-Earth3-CC and EC-Earth3-Veg simulate the climatology of precipitation and temperature? Does this affect their ability to simulate burned area in Africa?

**Reply:** We have added Figs. S5 and S6 to show the bias in temperature and precipitation climatology simulations of each CMIP6 models during and out of the fire seasons, respectively.

We have also added words after the paragraph in Sec. 3.2 "The significant underestimation of burned area in EC-Earth3 models in CMIP6 for Africa (around 1/5 and 1/10 of observations for NH and SH Africa, respectively, as shown in Fig. 3 and Table S1) is primarily due to limitations of the GlobFIRM fire model they employ as discussed in Sec. 4.2. However, climate simulation biases may also affect fire

simulations to some extent. During fire seasons, EC-Earth3 exhibits a cool bias in NH Africa, similar to other CMIP6 models except CESM2 (Figs. S6l and n). This cool bias may decrease fuel flammability due to reduced water evaporation from fuel, leading to underestimation of burned area. In contrast, SH Africa shows a warm bias (Figs. S6k and m), which tends to cause an overestimation of burned area. During fire seasons, EC-Earth3 models show no significant precipitation biases and do not have larger precipitation biases than other ESMs in Africa (Fig. S5). Outside of fire seasons, EC-Earth3 models exhibit distinct precipitation biases across Africa. In NH Africa, EC-Earth3 models show a dry bias (Figs. S5m and o), even though this dry bias is less pronounced than in CNRM-ESM2-1 (Fig. 5i), it may contribute to lower burned area estimates due to underestimated fuel load. In SH Africa, EC-Earth3 models display a wet bias, which potentially leads to higher burned area estimates due to overestimated fuel load.".

Page 11, Line 17: What are the possible reasons for models overestimating burned area in the South American savannas? Is this due to biases in simulated climate background or issues with the fire module?
**Reply:** We have added "possibly due to the underestimation of precipitation in this region during the fire seasons (Fig. S5)".

Figure 3: Please include the global average values of burned area from the three observational datasets in the figure.
**Reply:** Fig. 3 is designed to illustrate the spatial patterns of observed and simulated burned area, as well as provide global spatial correlation coefficients between simulations and observations. The global totals of burned area for observations and simulations across different models are already compared in Fig. 2a. To maintain clarity and avoid redundancy, we believe it's best to keep the global totals in Fig. 2a rather than repeating them in Fig. 3.

Figure 4: Please label the global average values of fire carbon emissions from the three observational datasets in the figure.
**Reply**: Fig. 3 shows the spatial pattern of observed and simulated fire carbon emissions and provides the global spatial correlation coefficients between simulations and observations. The global totals of fire carbon emissions are already shown in Fig. 2b.

Figure 5: Please indicate the data sources for the observed burned area and carbon emissions in the figure.
**Reply**: In Fig. 5 caption, we have added "The benchmarks are GFED5, FireCCI5.1 and MODIS C6 for burned area, and GFED4s, GFAS1.2, and FEER-G1.2 for fire carbon emissions".

Page 17, Line 2: Please provide some references for the observed decline in burned area.

**Reply**: The reference for this observed decline is indeed provided in Fig. 7, which is cited at the end of the sentence in question. To enhance clarity, we modified the sentence to "In recent decades, satellite-based products have revealed a significant decline in burned area and fire carbon emissions (dashed lines with shades in Fig. 7), but CMIP6 models do not capture this trend (bars in Fig. 7)"

Page 17, Lines 5-10: This section of discussion seems to lack corresponding figures. Please consider adding relevant illustrations.
**Reply**: Thank you for your suggestion. In this section, we provide specific numerical values for observed and simulated burned area trends in the three regions discussed. We believe these quantitative comparisons are clear and sufficient without additional figures.

Figure 10: How is the coefficient of variability (CV) of interannual variability defined?
**Reply**: We have added the definition of CV in the caption of Fig. 10 as "the standard deviation divided by the mean".

Page 20, Line 5: Why do models using the Li scheme show different errors in simulating the interannual variability of burned area and carbon emissions?
**Reply**: The different errors may be attributed to different simulations in climate and land physics as well as in vegetation structure and functions within CMIP6 coupled models using the Li scheme.

Page 20, Line 20: Are the errors in MPI models possibly due to poor simulation of relationships with ENSO and precipitation?
**Reply**: To explore the error sources in ENSO-fire carbon emission simulations, we have added new figures to compare the observed and simulated correlation between precipitation and fire carbon emissions (Fig. S13) and between ENSO and precipitation (Fig. S14).
We have also added our analyzed results in the text as "The failure of these models mainly stems from their poor simulations of the relationship between local precipitation and fire carbon emissions in Southeast Asia (i.e., mainly caused by fire model) (Fig. S13a) and are due to poor simulations of both the relationships between El Niño and precipitation and between precipitation and fire carbon emissions for eastern Africa and equatorial South America (Figs. S13b–c and S14b–c).".